# Supramolecularly directed rotary motion in a photoresponsive receptor

Sander J. Wezenberg [1] & Ben L. Feringa [1]

Stimuli-controlled motion at the molecular level has fascinated chemists already for several decades. Taking inspiration from the myriad of dynamic and machine-like functions in nature, a number of strategies have been developed to control motion in purely synthetic systems. Unidirectional rotary motion, such as is observed in ATP synthase and other motor proteins, remains highly challenging to achieve. Current artificial molecular motor systems rely on intrinsic asymmetry or a specific sequence of chemical transformations. Here, we present an alternative design in which the rotation is directed by a chiral guest molecule, which is able to bind non-covalently to a light-responsive receptor. It is demonstrated that the rotary direction is governed by the guest chirality and hence, can be selected and changed at will. This feature offers unique control of directional rotation and will prove highly important in the further development of molecular machinery.

[1] Stratingh Institute for Chemistry, University of Groningen, Nijenborgh 4, 9747 AG Groningen, The Netherlands. Correspondence and requests for materials should be addressed to S.J.W. (email: s.j.wezenberg@rug.nl)

In the advent of a possible new era in which nanoscale machinery is able to perform useful tasks in our daily lives[1], control of molecular motion is of fundamental importance. Inspired by the wealth of dynamic and machine-like functions in the biological as well as the macroscopic world, chemists have explored ways to create artificial molecular machines[2–11]. Well-known examples of such machines include rotaxane-based muscles[12, 13], elevators[14] and synthesizers[15, 16], azobenzene-derived photoswitchable tweezers[17, 18], and a nanocar comprising rotary molecular motors[19]. With respect to the basic types of motion (linear, rotary, oscillating and reciprocating), achieving directionality in a molecule's rotation, as is observed in biological systems such as ATPase, has drawn major attention and still remains a formidable challenge. Towards this goal, our group has developed chiral overcrowded alkenes[20–22], which undergo unidirectional rotation around their central double bond under the influence of light and thermal energy. Likewise, unidirectional rotation has been demonstrated in imines and hemo-thioindigos by the groups of Lehn[23] and Dube[24], respectively. Alternatively, the group of Kelly[25] and our group[26, 27] employed chemical

transformations to induce unidirectional rotation around a single carbon–carbon bond, whereas the group of Leigh demonstrated unidirectional motion of a small ring around a larger ring in mechanically interlocked structures[28–30]. It is worth mentioning that the design constraints for motors that are powered by light, which are governed by the Bose–Einstein relations for absorption and emission of photons, are very different than for motors that use chemical energy, which are governed by microscopic reversibility[31, 32].

In all natural and synthetic molecular motor systems, the information that governs the direction of rotation is embedded in the (pseudo-)asymmetry of the molecular structure or otherwise, a specific sequence of chemical transformations. Is that a pre-requisite? Or is it possible to transmit this information differently, for example, through interaction with molecules in the environment? Here we demonstrate that, by following the principles of light-driven rotary molecular motors[20–22] and supramolecular chirality transfer[33–35], a chiral guest molecule induces unidirectional rotation around the double bond in a photoswitchable receptor. As the rotary direction is determined by the chirality of

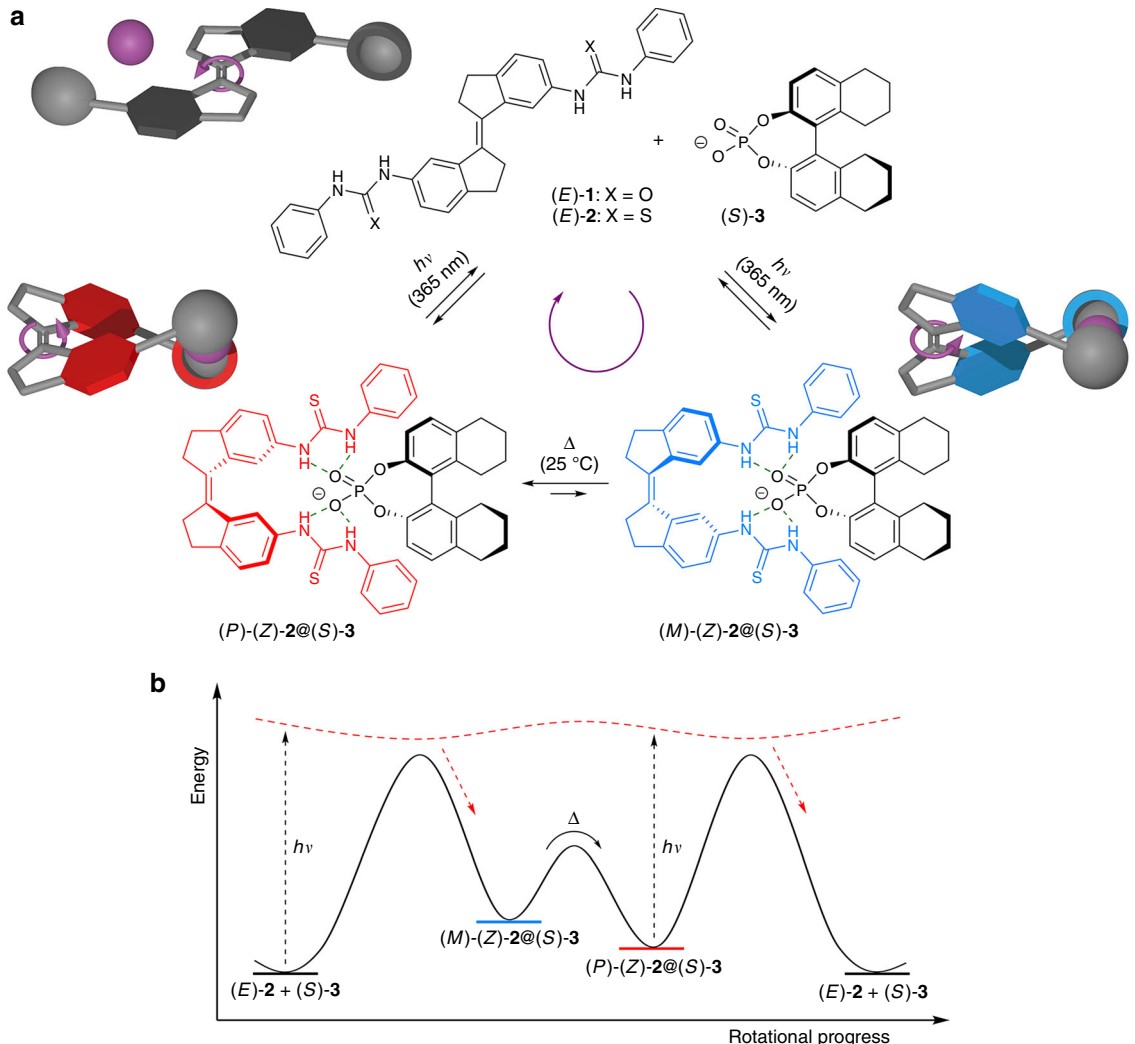

**Fig. 1** Chiral substrate induced unidirectional rotation in a photoswitchable receptor. **a** Scheme of photochemical and thermal isomerization steps showing that when starting from the planar (E)-isomer, e.g. bis-thiourea receptor (E)-**2**, irradiation with light produces a racemic mixture of helical (P)-(Z)-**2** and (M)-(Z)-**2**. Binding of a chiral guest molecule to the receptor, e.g. phosphate (S)-**3** favours formation of one of these helical (Z)-isomers. As a consequence, the backward photochemical isomerisation process, affording the (E)-isomer, takes place predominantly from the most favoured (Z)-isomer, i.e. (P)-(Z)-**2**@(S)-**3** resulting in a net unidirectional rotation. **b** Illustrative energy profile describing a full rotation. Note that (P)-(Z)-**2** and (M)-(Z)-**2** are produced with equal probability and that only formation of the first isomer gives rise to a rotation, while formation of the latter results in an oscillation

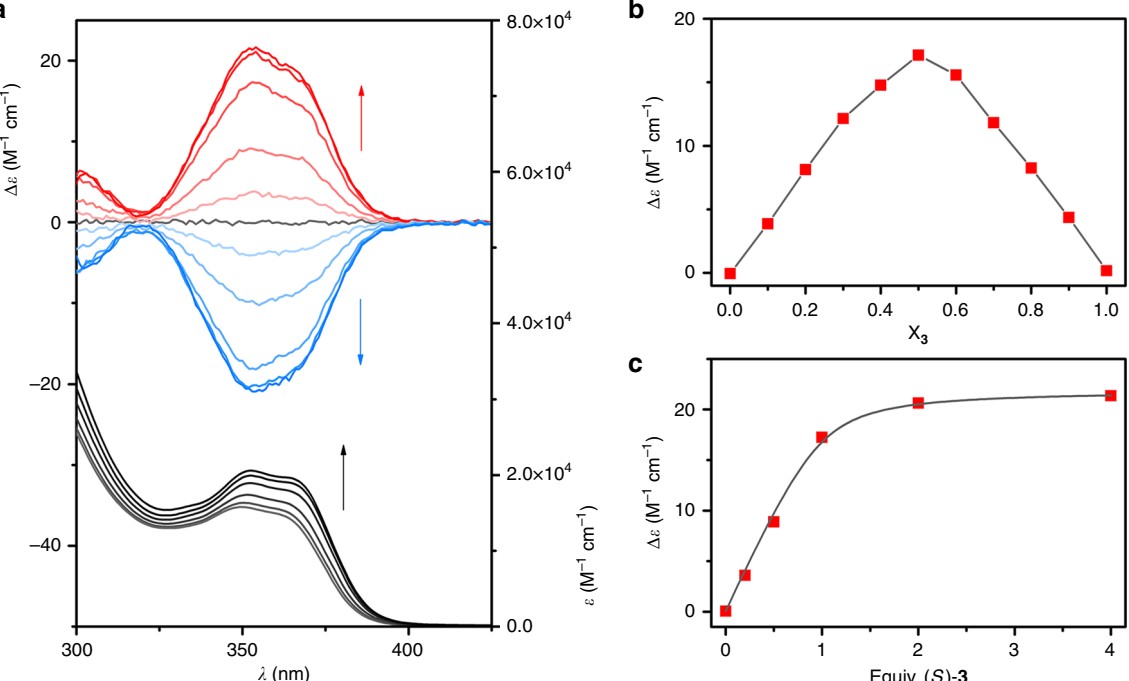

**Fig. 2** Supramolecular chirality induction. **a** CD (top) and UV–vis (bottom) spectra of (*Z*)-**2** (5.0 × 10$^{-4}$ M solution in CH$_2$Cl$_2$, 25 °C) in the presence of 0, 0.2, 0.5, 1.0, 2.0, and 4.0 equiv. of either [Bu$_4$N]$^+$ [(*S*)-**3**]$^-$ (CD, red line) or [Bu$_4$N]$^+$[(*R*)-**3**]$^-$ (CD, blue line). The appearance of a CD signal upon chiral phosphate addition reveals preferential formation of one of the helical forms of (*Z*)-**2**. **b** Job plot analysis obtained by mixing (*Z*)-**2** with [Bu$_4$N]$^+$[(*S*)-**3**]$^-$ (1.0 × 10$^{-3}$ M solutions in CH$_2$Cl$_2$, 25 °C) showing the CD absorption as function of the mole fraction of the latter (X$_3$). The maximum at 0.5 indicates a 1:1 binding stoichiometry. **c** Plot of the CD absorption versus the amount of equiv. of [Bu$_4$N]$^+$[(*S*)-**3**]$^-$ added and the calculated 1:1 binding isotherm ($K_a$ = 2.8 ± 0.3 × 10$^4$ M$^{-1}$)

the guest it can be selected and changed a posteriori. This type of development will prove crucial for increasing the level of control over nanoscale machinery.

## Results

**Receptor design and chirality induction.** Where various photoresponsive host–guest systems[36–38], containing different photoactive units, have been described, we based our design on a stiff-stilbene photoswitch forming the core of the receptor (Fig. 1a). This photoswitch is structurally rigid and can exist in *E* and *Z* configurations, which can be interconverted by light irradiation[39–41]. Density functional theory (DFT) modelling at the B3LYP/6-31G+(*d*,*p*) level of theory revealed that the (*Z*)-isomer adopts a helical conformation as steric crowding causes an out-of-plane distortion, whereas the (*E*)-isomer has a planar geometry (Supplementary Tables 1–3). Although stiff-stilbenes are known to have a very high activation barrier for thermal *E*–*Z* isomerisation[42], the barrier for going from the *P* to *M* helical form and vice versa of the (*Z*)-isomer was calculated to be only 16.7 kJ mol$^{-1}$ at room temperature (see Supplementary Fig. 5 and Supplementary Table 4 for details). This means that in solution the (*P*)-(*Z*)- and (*M*)-(*Z*)-isomers will rapidly interconvert and will be present in equal amounts (cf. racemic mixture). We anticipated that one of these isomers would be favoured over the other upon binding of a chiral substrate due to the formation of diastereomeric complexes with distinct stabilities. Where photoirradiation of the (*E*)-isomer will lead to the formation of either the (*P*)-(*Z*)- or (*M*)-(*Z*)-isomer with equal probability, the reverse photochemical reaction will then predominantly take place from the most favoured helical form. Hence, at the photostationary state, where the rates of forward and backward photochemical

isomerisation processes are identical given that both isomers absorb light at the irradiation wavelength, a net unidirectional rotation around the central double bond will occur. Similar to light-driven rotary molecular motors[20–22], this unidirectional rotation is ensured by the energy difference between diastereomeric forms (Fig. 1b). Transfer of chiral information by supramolecular means has been successfully applied in the past to induce a preferred helicity in helical polymers[34], and biaryl compounds[35], among others, and is here utilised to induce a unidirectional rotation in a molecule.

Urea and thiourea containing receptors are known to strongly interact with phosphate anions by hydrogen bonding in organic solvents[43–45]. In a recent study we found that the (*Z*)-isomer of bis-urea **1** (Fig. 1) effectively binds dihydrogen phosphate in the competitive DMSO/0.5% H$_2$O solvent mixture ($K_a$ = 2.02 × 10$^3$ M$^{-1}$)[46]. This receptor could be switched successfully between (*E*)- and (*Z*)-isomers using light, which brings about a large change in anion binding affinity. For more bulky phosphate anions, a decrease in the overall binding strength is expected[47]. Therefore, in our current studies, the use of a solvent that competes less with hydrogen bonding is desired. Unfortunately, compound **1** turned out to be very poorly soluble in other solvents than DMSO, but the related bis-thiourea analogue **2** (Fig. 1) nonetheless proved to be fairly soluble in CH$_2$Cl$_2$. Both (*E*)- and (*Z*)-isomers of receptor **2** were obtained in high yield (89% and 76%, respectively) by following a similar procedure as for bis-urea **1**[46], i.e. by reaction of the corresponding diamine precursors with phenyl isothiocyanate. The enantiomers of phosphoric acid **3** were synthesised starting from optically pure H8-BINOL and the respective phosphate salts [Bu$_4$N]$^+$[(*S*)-**3**]$^-$ and [Bu$_4$N]$^+$[(*R*)-**3**]$^-$ were accessed through treatment with tetrabutylammonium hydroxide. The use of H8-BINOL was preferred over BINOL because the

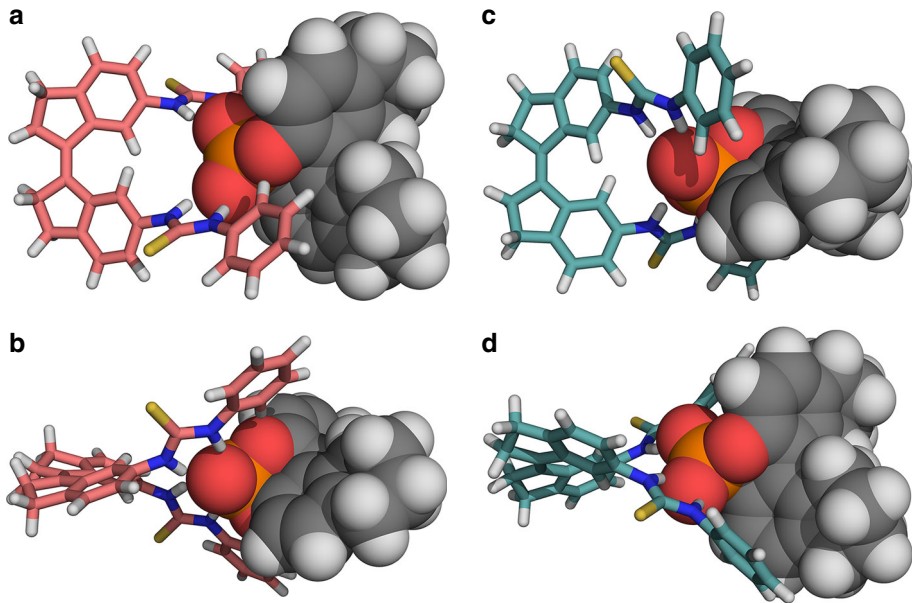

**Fig. 3** DFT optimised geometries. **a**, **b** Top (**a**) and side (**b**) view of the complex (*P*)-(*Z*)-**2** ⊃ (*S*)-**3**. **c**, **d** Top (**c**) and side (**d**) view of the complex (*M*)-(*Z*)-**2** ⊃ (*S*)-**3**. The bis-thiourea receptor (*Z*)-**2** is depicted as a stick model [C, pink for (*P*)-isomer, light blue for (*M*)-isomer; N, blue, S, yellow, H, white] and the phosphate anion (*S*)-**3** as a space-filling model (C, grey; O, red; P, orange; H, white). The structures were optimised at the B3LYP-6-31G++(*d*,*p*) level of theory using an IEFPCM, CH$_2$Cl$_2$ solvation model

absorption spectrum of the latter overlaps with that of stiff-stilbene and therefore, would complicate circular dichroism (CD) studies (vide infra).

It was then probed whether one of the helical isomers of (*Z*)-**2** formed preferentially upon addition of the H8-BINOL-derived phosphate salts. Anion complexation became evident from [1]H NMR studies (Supplementary Fig. 1), which showed large downfield displacements of the signals that belong to the thiourea protons upon [Bu$_4$N]$^+$[(*S*)-**3**]$^-$ addition to the receptor in CD$_2$Cl$_2$. At room temperature, the [1]H NMR signals of the two possible diastereomeric complexes [i.e. (*P*)-(*Z*)-**2** ⊃ (*S*)-**3** and (*M*)-(*Z*)-**2** ⊃ (*S*)-**3**] could not be distinguished pointing to fast interconversion on the NMR time scale. Upon decreasing the temperature, however, the signals first broadened (coalescence around −20 °C) and subsequently two separate sets of signals could be identified (below −55 °C). The integrated average intensities of the thiourea signals at low temperature revealed a ratio of approximately 1:10 between diastereomeric complexes, which implicates an efficient chirality induction process. Noteworthy, when the phosphate anion was absent, (*Z*)-**2** was found to aggregate beyond a concentration of 1 mM (Supplementary Fig. 2).

In the CD spectrum, stepwise addition of the (*S*)-H8-BINOL-derived phosphate anion to (*Z*)-**2** in CH$_2$Cl$_2$ resulted in the appearance of a positive band ($\lambda_{max}$ ~355 nm), while addition of the (*R*)-enantiomer led to a signal with opposite sign (Fig. 2a). Furthermore, the overall UV–vis absorption ($\lambda_{max}$ ~355 nm) increased during these additions. As the used phosphate salt does not absorb light above $\lambda$ = 300 nm (Supplementary Fig. 3), the CD spectral changes can be fully ascribed to preferential formation of one of the helical isomers of (*Z*)-**2**. Job plot analysis, performed by plotting the CD absorption against the mole fraction of [Bu$_4$N]$^+$[(*S*)-**3**]$^-$, confirmed the anticipated 1:1 receptor/phosphate stoichiometry (Fig. 2b). When the data obtained upon stepwise phosphate addition were fitted to a 1:1 binding model using BindFit (Supramolecular.org, http://supramolecular.org/), an average stability constant of $K_a$ = 2.8 ± 0.3 × 10$^4$ M$^{-1}$ was calculated (Fig. 2c). Importantly,

addition of [Bu$_4$N]$^+$[(*S*)-**3**]$^-$ to (*E*)-**2**, under the same experimental conditions, did not lead to the induction of any significant CD absorption band (Supplementary Fig. 4).

**Structural characterisation.** DFT calculations were carried out to gain insight into which of the possible diastereomeric complexes is the most stable one. The geometries of (*P*)-(*Z*)-**2** ⊃ (*S*)-**3** and (*M*)-(*Z*)-**2** ⊃ (*S*)-**3** were optimised at the B3LYP/6-31G++(*d*,*p*) level of theory, using an IEFPCM CH$_2$Cl$_2$ solvation model (Fig. 3a–d and Supplementary Tables 5, 6 for details). Distinct binding modes of the H8-BINOL-derived phosphate anion with either the (*P*)- or (*M*)-helical form of bis-thiourea were observed. In the structure of (*P*)-(*Z*)-**2** ⊃ (*S*)-**3**, a weak C–H⋯O interaction (C⋯O distance: 3.58 Å; C–H⋯O angle: 167°) between the phenyl substituents of the receptor and the H8-BINOL oxygens can be identified, beside the N–H⋯O hydrogen bonding interactions (N⋯O distances: 2.86–2.87 Å; N–H⋯O angles: 155–159°). Thiourea hydrogen bonding is highly similar in (*M*)-(*Z*)-**2** ⊃ (*S*)-**3** (N⋯O distances: 2.85–2.90 Å; N–H⋯O angles: 155–159°), but additional C–H⋯O bonding is not found. Furthermore, steric interactions between the phenyl substituents and the H8-BINOL moiety are larger in (*M*)-(*Z*)-**2** ⊃ (*S*)-**3** than in (*P*)-(*Z*)-**2** ⊃ (*S*)-**3** (shortest H⋯H contact: 3.18 and 3.95 Å, respectively). In agreement with these structural observations, the latter diastereomeric complex [i.e. (*P*)-(*Z*)-**2** ⊃ (*S*)-**3**] was found to be 5.6 kJ mol$^{-1}$ lower in Gibbs free energy.

Based upon the finding that (*P*)-(*Z*)-**2** ⊃ (*S*)-**3** is the lowest energy complex, it is concluded that the positive CD signal (vide supra) stems from the (*P*)-helical isomer of bis-thiourea **2** and vice versa. The theoretical CD spectrum of (*P*)-(*Z*)-**2** ⊃ (*S*)-**3**, calculated using time-dependent DFT at the same level of theory as the geometry optimisations, compares well with the experimentally obtained spectrum (Supplementary Fig. 7). Binding of the (*S*)-H8-BINOL-derived phosphate anion to (*Z*)-**2** thus induces formation of the (*P*)-helical isomer, while binding of the (*R*)-enantiomer favours the (*M*)-helical isomer.

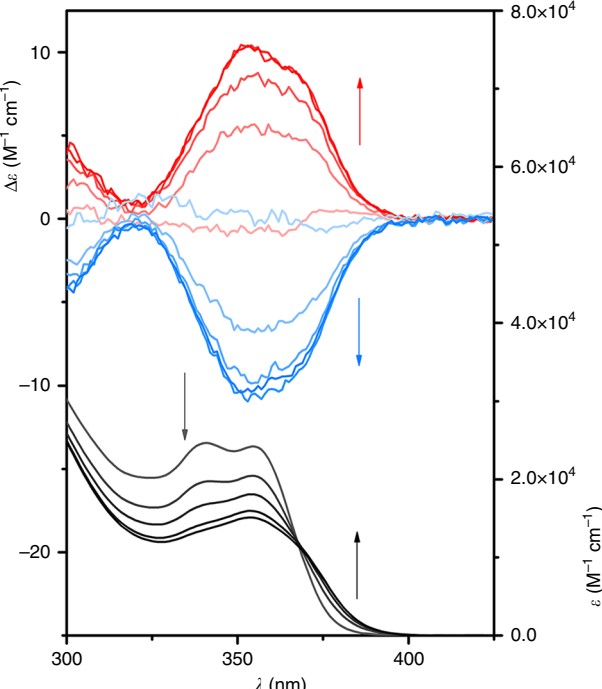

**Fig. 4** Photochemically induced isomerisation. CD (top) and UV–vis (bottom) spectral changes upon irradiation for 0, 10, 20, 40 and 60 s with $\lambda_{max} = 365$ nm light of a mixture of (*E*)-**2** ($5.0 \times 10^{-4}$ M solution in $CH_2Cl_2$, 25 °C) and either 2 equiv. $[Bu_4N]^+[(S)$-**3**]$^-$ (CD, red line) or $[Bu_4N]^+[(R)$-**3**]$^-$ (CD, blue line). The appearance of a CD signal upon irradiation is caused by photogenerated formation of (*Z*)-**2** and simultaneous induction of chirality by the chiral phosphate anion

**Photochemical isomerisation**. A mixture of (*E*)-**2** and either enantiomer of the phosphate anion gave a virtually silent CD spectrum (Fig. 4). However, when the sample was irradiated with 365 nm light, a CD signal gradually appeared. The signal was positive for the sample containing $[Bu_4N]^+[(S)$-**3**]$^-$ and negative for the one with $[Bu_4N]^+[(R)$-**3**]$^-$, in agreement with the chirality induction experiments (vide supra). At the same time, the absorption maxima in the UV–vis spectrum ($\lambda_{max} = \sim340$ and $\sim 355$ nm) decreased and the absorption band shifted bathochromically, which is similar to what has been reported for *E–Z* isomerisation of compound **1**[46]. The appearance of the CD absorption bands upon irradiation must therefore originate from formation of the (*Z*)-isomer and concomitant induction of helical chirality by binding of the chiral phosphate anion. The samples were irradiated until no further changes were noted, i.e. the photostationary state (PSS) was reached. The clear isosbestic point observed at $\lambda = 365$ nm reveals that this photoinduced isomerisation is a unimolecular process. The PSS ratios in the presence and absence of $[Bu_4N]^+[(S)$-**3**]$^-$ were determined by $^1$H NMR spectroscopy (Supplementary Figs. 8, 9) and were found to be 24:76 and 42:58 (*E*/*Z*), respectively.

The quantum yield for this $E \longrightarrow Z$ photoisomerisation process was determined by comparing the rate of formation of (*Z*)-**2** to the rate of formation of $Fe^{2+}$ ions from potassium ferrioxalate under identical conditions (Supplementary Fig. 10). Starting with solutions of (*E*)-**2** in the presence and absence of $[Bu_4N]^+[(S)$-**3**]$^-$, at concentrations high enough to absorb all incoming light, the increase in concentration of (*Z*)-**2** was monitored over time by following the absorption increase at $\lambda = 380$ nm (Supplementary Figs. 11–13). It was found that the quantum yield for the forward isomerisation process ($\Phi_{E \longrightarrow Z} = 18.2\% \pm 1.5\%$) is not significantly affected by the presence of the phosphate anion

($\Phi_{E \longrightarrow Z} = 20.1\% \pm 1.6\%$). As the molar absorptivity ($\varepsilon$) of each isomer is similar at the irradiation wavelength (cf. isosbestic point), the higher PSS ratio found in the presence of $[Bu_4N]^+[(S)$-**3**]$^-$ should be ascribed to a lower quantum yield for the backward *Z–E* isomerisation process (note that: $\Phi_{Z \longrightarrow E} = \Phi_{E \longrightarrow Z}$ $\varepsilon_E$ $n_E/\varepsilon_Z$ $n_Z$). Hence, H8-BINOL-derived phosphate binding to the (*Z*)-isomer seems to influence the photochemical quantum yield. The underlying cause for this observation, however, is still unclear and requires further investigation.

## Discussion

The combination of experiments that we have performed is consistent with supramolecularly directed rotation around the central double bond of the photoswitchable receptor. Once the photostationary state is reached, and irradiation is continued, the (*E*)- and (*Z*)-isomers have equal rates of formation since they both absorb $\lambda = 365$ nm light. Isomerisation of (*E*)-**2** leads to formation of the rapidly interconvertible (*P*)-(*Z*)-**2** and (*M*)-(*Z*)-**2** with equal probability. As the binding of a chiral phosphate anion, e.g. (*S*)-**3**, then induces preferential formation of one of the helical isomers, i.e. (*M*)-(*Z*)-**2** ⊃ (*S*)-**3** will convert to (*P*)-(*Z*)-**2** ⊃ (*S*)-**3** by a helicity inversion, the reverse isomerisation pathway takes place predominantly from the latter form. Resultantly, a net unidirectional rotation occurs. It should be noted that the operation principle of this system is similar to that of light-driven rotary molecular motors[20–22] and that it is the generation of the higher energy (metastable) diastereomeric complex in the photochemical step that drives unidirectional rotation (Fig. 1b).

In our system the rotary motion is directed by an external substrate rather than (pseudo-)asymmetry in the molecular design or a specific sequence of chemical transformations. This unique approach will provide new views on how to control motion on the nanoscale with the ultimate goal of bringing nanoscale machinery to a higher level of complexity and sophistication.

## Methods

**(*E*)-1,1'-(2,2',3,3'-tetrahydro-[1,1'-biindenylidene]-6,6'-diyl)bis(3-phenylthiourea) [(*E*)-2]**. Phenyl isothiocyanate (25 µL, 0.21 mmol) was added to (*E*)-2,2',3,3'-tetrahydro-(1,1'-biindenylidene)-6,6'-diamine (25 mg, 0.10 mmol) in THF (1 mL) under a $N_2$ atmosphere. The mixture was stirred for 16 h, after which the white precipitate was filtered off, washed with THF and dried in vacuo to afford (*E*)-**2**·THF (51 mg, 89%) as a white solid: m.p. 188.7–190.4 °C; $^1$H NMR (400 MHz, DMSO-$d_6$): 9.81 (s, 2H; NH), 9.77 (s, 2 H; NH), 7.83 (s, 2H; NH), 7.50 (d, *J* = 7.6 Hz, 4H; arom. H), 7.37–7.25 (m, 8H; arom. H), 7.13 (t, *J* = 7.2 Hz, 2H; arom. H), 3.60 (m, 4H; THF), 3.15–3.01 (m, 8H; $CH_2$), 1.76 (m, 4H; THF); $^{13}$C NMR (100 MHz, DMSO-$d_6$): 179.7, 143.1, 142.5, 139.5, 137.9, 135.0, 128.4, 124.5, 124.3, 123.6, 123.0, 120.0, 67.0 (THF), 31.6, 30.0, 25.1 (THF); HRMS (ESI) *m/z*: 533.1819 ([M+H]$^+$, calcd for $C_{32}H_{29}N_4S_2^+$: 533.1828).

**(*Z*)-1,1'-(2,2',3,3'-tetrahydro-[1,1'-biindenylidene]-6,6'-diyl)bis(3-phenylthiourea) [(*Z*)-2]**. Phenyl isothiocyanate (36 µL, 0.30 mmol) was added to (*Z*)-2,2',3,3'-tetrahydro-(1,1'-biindenylidene)-6,6'-diamine (39 mg, 0.15 mmol) in $CH_2Cl_2$ (2 mL) under a $N_2$ atmosphere. The solution was stirred for 16 h, after which the white precipitate was filtered off, washed with $CH_2Cl_2$ and air-dried to afford (*Z*)-**2** (61 mg, 76%) as a white solid: m.p. 184 °C (decomp); $^1$H NMR (400 MHz, DMSO-$d_6$): 9.68 (s, 2H; NH), 9.55 (s, 2H; NH), 8.15 (s, 2H; arom. H), 7.48 (d, *J* = 7.6 Hz, 4H; arom. H), 7.30–7.22 (m, 8H; arom. H), 7.07 (t, *J* = 7.2 Hz, 2H; arom. H), 2.96–2.77 (m, 8H; $CH_2$); $^{13}$C NMR (100 MHz, DMSO-$d_6$): 179.5, 144.4, 139.9, 139.4, 137.4, 134.8, 128.4, 124.9, 124.2, 123.5, 123.4, 119.0, 34.8, 29.6; HRMS (ESI) *m/z*: 533.1820 ([M+H]$^+$, calcd for $C_{32}H_{29}N_4S_2^+$: 533.1828).

**(*S*)-5,5',6,6',7,7',8,8'-octahydro-1,10-binaphthyl-2,2'-diyl hydrogen phosphate [(*S*)-3]**. Phosphoryl chloride (74 µL, 0.79 mmol) was added slowly to (*S*)-5,5',6,6',7,7',8,8'-octahydro-1,1'-bi-2-naphthol (147 mg, 0.50 mmol) in pyridine (3 mL) under a $N_2$ atmosphere. The solution was stirred for 16 h, treated with $H_2O$ (3 mL) and stirred for a further 2 h. Then the solution was poured into $CH_2Cl_2$ (15 mL) and the resulting mixture was stirred for 1 h and subsequently acidified by the addition of concentrated HCl (0.5 mL). The organic layer was separated, washed with 2 M aqueous HCl (2 × 10 mL), concentrated and dried in vacuo. The product was purified by FC ($SiO_2$, 10% MeOH in $CH_2Cl_2$), redissolved in $CH_2Cl_2$ (20 mL)

and washed with 2 M aqueous HCl (3 × 10 mL) to remove residual pyridine. The volume of the organic layer was reduced to 5 mL and pentane was added. The resulting white precipitate was filtered off, washed with pentane and air-dried to afford (S)-3 (137 mg, 77%) as a white solid: m.p. 314 °C (decomp); $[\alpha]_D^{20} = +233$ ($c = 1.0$ in EtOH); [1]H NMR (400 MHz, DMSO-$d_6$): 7.16 (d, $J = 8.2$ Hz, 2H), 7.00 (d, $J = 8.2$ Hz, 2H), 2.87–2.71 (m, 4H), 2.69–2.59 (m, 2H), 2.19–2.09 (m, 2H), 1.80–1.68 (m, 6H), 1.52–1.42 (m, 2H); [13]C NMR (100 MHz, DMSO-$d_6$): 146.6, 137.5, 134.5, 129.7, 126.0, 118.4, 28.4, 27.3, 22.0, 21.9; [31]P NMR (162 MHz, DMSO-$d_6$): 1.18; HRMS (ESI) $m/z$: 357.1229 ([M+H]$^+$, calcd for $C_{20}H_{22}O_4P^+$: 357.1250). The enantiomer (R)-3 was obtained following a similar procedure: $[\alpha]_D^{20} = -240.8$ ($c = 1.0$ in EtOH);[48] -249.9 ($c = 1.0$ in EtOH).

**Tetrabutylammonium (S)-5,5′,6,6′,7,7′,8,8′-Octahydro-1,10-binaphthyl-2,2′-diyl phosphate: {[NBu$_4$]$^+$[(S)-3]$^−$}.** Tetrabutylammonium hydroxide 30-hydrate (160 mg, 0.20 mmol) and compound (S)-3 (72 mg, 0.20 mmol) were dissolved in MeOH (5 mL). The solution was stirred for 3 h and concentrated, which was followed by repetitive solution/evaporation cycles using first MeOH (2×) and then CH$_2$Cl$_2$ (3×). The concentrate was dried in vacuo to afford [Bu$_4$N]$^+$[(S)-3]$^−$ (119 mg, 99%) as an off-white solid: m.p. 81.6–83.1 °C; $[\alpha]_D^{20} = +132$ ($c = 0.2$ in CHCl$_3$); [1]H NMR (400 MHz, DMSO-$d_6$): 6.99 (d, $J = 8.2$ Hz, 2H), 6.76 (d, $J = 8.2$ Hz, 2H), 3.16 (br. $t$, $J = 8.0$ Hz, 8H), 2.82–2.56 (m, 6H), 2.16–2.07 (m, 2H), 1.78–1.42 (br. m, 16H), 1.37–1.24 (m, 8H), 0.93 (t, $J = 7.3$ Hz, 12H); [13]C NMR (100 MHz, DMSO-$d_6$): 149.6, 136.3, 131.7, 128.4, 127.6, 119.0, 57.5, 28.5, 27.4, 23.1, 22.3, 22.2, 19.2, 13.5; HRMS (ESI) $m/z$: 355.1105 ([M–H]$^−$, calcd for $C_{20}H_{20}O_4P^−$: 355.1105). The enantiomer [NBu$_4$]$^+$[(R)-3]$^−$ was obtained following a similar procedure: $[\alpha]_D^{20} = -136$ ($c = 0.1$ in CHCl$_3$).

**NMR addition and irradiation experiments**. The desired amount of phosphate salt [NBu$_4$]$^+$ [3]$^−$ was dissolved in a 2 mM solution of bis-thiourea 2 in CD$_2$Cl$_2$ (degassed prior to irradiation experiments). Irradiation was performed with a Thorlab model M365FP1 high-power LED (15.5 mW) coupled to a 600 μm optical fibre, which guided the light into the NMR tube inside the spectrometer[49].

**CD/UV–vis addition and irradiation experiments**. A $5 \times 10^{-3}$ M solution of phosphate salt [NBu$_4$]$^+$[3]$^−$ in CH$_2$Cl$_2$ (degassed prior to irradiation experiments) containing $5 \times 10^{-4}$ M bis-thiourea 2 was added to 200 μL of a $5 \times 10^{-4}$ M solution of bis-thiourea 2 in a 1 mm quartz cuvette (50 μL = 2 equiv.). Irradiation was carried out using a Thorlab model M365F1 high-power LED (4.1 mW) positioned at a distance of 1 cm from the cuvette.

**Data availability**. The data associated with the reported findings are available in the manuscript or the Supplementary Information. Other related data are available from the corresponding author upon request.

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

## Acknowledgements

This work was financially supported by The Netherlands Organization for Scientific Research (NWO-CW, Veni Grant No. 722.014.006 to S.J.W.), the Ministry of Education, Culture and Science (Gravitation Program No. 024.001.035), and the European Research Council (Advanced Investigator Grant No. 694345 to B.L.F.) We thank Pieter van der Meulen for assistance with NMR irradiation experiments.

## Author contributions

S.J.W. and B.L.F. conceived the project. S.J.W. carried out the experimental work and the theoretical calculations. S.J.W. analysed the results and wrote the manuscript with assistance from B.L.F.

## Additional information

**Competing interests:** The authors declare no competing interests.

