## [Peer Review File · Nature Communications]

Reviewers' comments:

Reviewer #1 (Remarks to the Author):

Overall this is a very interesting manuscript that illustrates how chirality can be used to control the direction of rotation of a synthetic molecular rotor. My only caution is that the system in the present paper involves driving by light. The design constraints for light driven motors are very different than for rotors and motors driven by catalysis of a chemical reaction such as ATP hydrolysis. This has been pointed out in the recent literature in several places (e.g., Pezzato et al. Chem. Soc. Rev. 46:5467-5680, 2017). I think that it is very important that the authors be abundantly clear that design lessons from light driven systems that are governed by the Einstein relations for absorption and emission of photons may not (even probably will not) transfer to systems that are intended to use energy provided by catalysis of a non-equilibrium reaction, be it ATP hydrolysis or conversion of FMOC-Cl to Dibenzofulvene, all of which are governed by the principle of microscopic reversibility.

I recommend publication after the authors have had the opportunity to consider these comments, and the comments of the other reviewers.

Reviewer #2 (Remarks to the Author):

Wezenberg and Feringa demonstrated in this work supramolecularly-directed, unidirectional rotation by adopting combination of thiourea-containing photoswitchable receptor 2 and chiral phosphoric acid 3. The system is cleverly designed and the supramolecularly-directed, unidirectional rotation is unprecedented. All the compounds and complexes are characterized well. In particular, the helicity of the complex was established by NMR and CD spectra along with theoretical calculations. The work is of broad-spectrum chemical interest and will garner much interest from the readership of this journal. Therefore, I recommend this manuscript for publication in this journal, after the authors address the following minor points.

1. Upon irradiation, the system reached photostationary state within 40 s, and irradiation was continued until 60 s. What happened after 60 s?

2. The association constant between 2 and 3 was determined to be 2.8×10^4 by using CD data. I encourage the authors to add a margin of error to the value. Could the same value be obtained by using absorption spectra where 3 does not absorb light?

3. The chiral phosphoric acid 3 derived from H8-BINOL worked very well. Have the authors tried other chiral phosphoric acid derived from BINOL or tartaric acid?

4. As the authors say, a net unidirectional rotation occurs since the Z-to-E isomerization takes place predominantly from the major complex at the photostationary state. Of course, the isomerization takes place from the minor complex, too. Could the net unidirectional rotation rate be calculated?

Reviewer #3 (Remarks to the Author):

The rotary system consists of a molecule that can be switched reversibly between chiral and achiral forms by irradiation (and can be held in a photostationary state with equal rates of transformation in

both directions). Addition of a permanently chiral guest molecule induces relaxation of the chiral form of the photoswitch to a preferred chiral conformation. In the presence of this guest, in the photostationary state, there is thus a continuous net rotation of the chiral photoswitch population (as a result of interaction with the chiral guest) which is continually undone by photoconversion to the achiral form.

No photoswitch molecule rotates continuously. Each oscillates stochastically between chiral forms, relaxing to a preferred chirality. Optical excitation continuously destroys the chirality of the photoswitch population, which is restored through interaction with the guest. I am not sure that this is much of a rotary machine.

The work is original and carried out to a high standard. As the discussion above shows, my only reservation is that I am not sure that it represents an important advance.

We are pleased with the very positive response to our manuscript entitled “Supramolecularly directed rotary motion in a photoresponsive receptor”. We appreciate the comments and suggestions made by the three reviewers and we have revised our manuscript accordingly. In particular, to address the remarks made by reviewer 1 and 3, we have provided a more detailed explanation on what is the driving force for rotation in our system. Below you find a point-by-point response to all comments.

Reviewer #1:

Overall this is a very interesting manuscript that illustrates how chirality can be used to control the direction of rotation of a synthetic molecular rotor. My only caution is that the system in the present paper involves driving by light. The design constraints for light driven motors are very different than for rotors and motors driven by catalysis of a chemical reaction such as ATP hydrolysis. This has been pointed out in the recent literature in several places (e.g., Pezzato et al. Chem. Soc. Rev. 46:5467-5680, 2017). I think that it is very important that the authors be abundantly clear that design lessons from light driven systems that are governed by the Einstein relations for absorption and emission of photons may not (even probably will not) transfer to systems that are intended to use energy provided by catalysis of a non-equilibrium reaction, be it ATP hydrolysis or conversion of Fmoc-Cl to Dibenzofulvene, all of which are governed by the principle of microscopic reversibility.

I recommend publication after the authors have had the opportunity to consider these comments, and the comments of the other reviewers.

The presented system is indeed driven by light. As mentioned by the reviewer, it has been very nicely demonstrated by Astumian et al. that light-driven motors follow very different design principles than chemically-driven motors. In fact, the operating principle of the currently presented system is highly similar to that of the light-driven rotary molecular motors that we previously developed in our group.

That is, light irradiation results in the formation of a “metastable” diastereomeric complex (although in this case only for 50%), which can equilibrate to a more stable complex *via* a helicity inversion. We felt that this might not have been completely clear in the first version of our manuscript and we have therefore added an energy profile as Fig. 1b to the main text. Furthermore, we have made reference to the work by Astumian in the introduction and we have emphasized the similarity to light-driven rotary molecular motors in the introduction and discussion section.

Reviewer #2:

Wezenberg and Feringa demonstrated in this work supramolecularly-directed, unidirectional rotation by adopting combination of thiourea-containing photoswitchable receptor 2 and chiral

phosphoric acid 3. The system is cleverly designed and the supramolecularly-directed, unidirectional rotation is unprecedented. All the compounds and complexes are characterized well. In particular, the helicity of the complex was established by NMR and CD spectra along with theoretical calculations. The work is of broad-spectrum chemical interest and will garner much interest from the readership of this journal. Therefore, I recommend this manuscript for publication in this journal, after the authors address the following minor points.

1. Upon irradiation, the system reached photostationary state within 40 s, and irradiation was continued until 60 s. What happened after 60 s?

The spectral changes observed after 40 s irradiation were minimal indicating that the photostationary state had been reached. When irradiation is continued after 60 s, the spectra will not change further (although eventually the absorption will decrease due to some degradation of the receptor).

2. The association constant between 2 and 3 was determined to be 2.8×10^4 by using CD data. I encourage the authors to add a margin of error to the value. Could the same value be obtained by using absorption spectra where 3 does not absorb light?

An error margin, obtained from BindFit, has been added. Following the suggestion of the reviewer, we have additionally used the UV-Vis absorption spectra for data fitting to the same 1:1 binding model using BindFit which gave a 3-fold lower constant ($K_a = 8.6 \times 10^3 \text{ M}^{-1}$) than the value obtained using the

CD data. However, we choose to rely on the CD spectra as they allow for a clear distinction between the 1:1 complex ($\Delta\epsilon = 20 \text{ M}^{-1} \text{ cm}^{-1}$) and the unbound receptor ($\Delta\epsilon = 0$) species. In the UV-Vis absorption spectra, some minor 2:1 complex formation (which may be present at higher equivalents) could cause

an absorption increase, leading to a lower estimation of the association constant.

3. The chiral phosphoric acid 3 derived from H8-BINOL worked very well. Have the authors tried other chiral phosphoric acid derived from BINOL or tartaric acid?

We have tried BINOL phosphoric acid, but its absorption spectrum overlaps with that of the bis-thiourea receptor and therefore, complicates the chirality transfer studies using CD spectroscopy. We have added this information to the main text on page 4. Based on related studies in our group, we presume that tartaric acid is not bulky enough to be an efficient chirality inducer in this system.

4. As the authors say, a net unidirectional rotation occurs since the Z-to-E isomerization takes place predominantly from the major complex at the photostationary state. Of course, the isomerization takes place from the minor complex, too. Could the net unidirectional rotation rate be calculated?

Photoisomerization can indeed take place from both major and minor diastereomeric complexes. In theory it should be possible to calculate a maximum rotation rate by assuming that the rate-

determining step, given an infinite photon flux, would be the thermal conversion from the higher energy complex [i.e. (*M*)-(*Z*)-**2**@(*S*)-**3**], generated upon irradiation, to the lower energy complex [i.e. (*P*)-(*Z*)-**2**@(*S*)-**3**]. We have therefore considered to determine this interconversion rate by EXSY NMR, however, as shown in Supplementary Fig. 1, the exchange rate is too fast. We hope to determine such a rate in the future for designs that have higher helicity inversion barriers than the current system.

Reviewer #3:

The rotary system consists of a molecule that can be switched reversibly between chiral and achiral forms by irradiation (and can be held in a photostationary state with equal rates of transformation in both directions). Addition of a permanently chiral guest molecule induces relaxation of the chiral form of the photoswitch to a preferred chiral conformation. In the presence of this guest, in the photostationary state, there is thus a continuous net rotation of the chiral photoswitch population (as a result of interaction with the chiral guest) which is continually undone by photoconversion to the achiral form.

No photoswitch molecule rotates continuously. Each oscillates stochastically between chiral forms, relaxing to a preferred chirality. Optical excitation continuously destroys the chirality of the photoswitch population, which is restored through interaction with the guest. I am not sure that this is much of a rotary machine.

The work is original and carried out to a high standard. As the discussion above shows, my only reservation is that I am not sure that it represents an important advance.

We are pleased that the reviewer recognizes the originality and quality of our work. However, it seems that this reviewer has not recognized the driving force for a net unidirectional rotation in our system. An oscillation will occur when *E-Z* photoisomerization affords the lower energy diastereomer and *Z-E* isomerization takes place from the same isomer. Nevertheless, with equal probability, the higher energy diastereoisomer is obtained after irradiation, which is subsequently converted thermally to the lower energy diastereomer. When *Z-E* isomerization takes place from the latter form, a full rotation is achieved. We are confident that by the addition of the energy profile and the caption in Fig. 1b, the operation mode of our system is fully clear to the reader. Furthermore, being the first example in which a rotary motion in a molecule can be controlled by an external substrate, we believe that this work represents a very important advance in the field of molecular machines.

We hope that the manuscript is now in good shape to be published. In case any issues remain, please let me know.

With best regards, Sander Wezenberg

REVIEWERS' COMMENTS:

Reviewer #2 (Remarks to the Author):

I have examined the revised manuscript carefully and have found that the authors have completely addressed the comments raised by the three reviewers. I am now convinced that this revised version is worth publication in this journal as is.